
# Impact of frozen soil processes on soil thermal characteristics at seasonal to decadal scales over the Tibetan Plateau and North China

Qian Li[1] , Yongkang Xue[2,3],    Ye Liu[2]

[1]Institute of Atmospheric Physics, Chinese Academy of Sciences, Beijing, 100029, China.

[2]Department of Geography, University of California Los Angeles (UCLA), CA, 90095-1524, USA.

[3]Department of Atmospheric and Oceanic Sciences, UCLA, CA, 90095-1524, USA.

*Correspondence to*: Qian Li (qian@mail.iap.ac.cn)

**Abstract.** Frozen soil processes are of great importance in controlling surface water and energy balances during the cold season and in cold regions. Over recent decades, considerable frozen soil degradation and surface soil warming have been reported over the Tibetan Plateau and North China, but most land surface models have difficulty in capturing the freeze-thaw cycle and few validations focus on the effects of frozen soil processes on soil thermal characteristics in these regions. This paper addresses these issues by introducing a physically more

realistic and computationally more stable and efficient frozen soil module (FSM) into a land surface model—the third-generation Simplified Simple Biosphere model (SSiB3-FSM). To overcome the difficulties in achieving stable numerical solutions for frozen soil, a new semi-implicit scheme and a physics-based freezing-thawing scheme were applied to solve the governing equations. The performance of this model, as well as the effects of frozen soil process on the soil temperature profile and soil thermal characteristics, were investigated over the

Tibetan Plateau and North China using observation and models. Results show that the SSiB3 model with the FSM produces more realistic soil temperature profile and its seasonal variation than that without FSM during the freezing and thawing periods. The freezing process in soil delays the winter cooling, while the thawing process delays the summer warming. The time lag and amplitude damping of temperature become more pronounced with increasing depth. These processes are well simulated in SSiB3-FSM. The freeze-thaw processes could increase

the simulated phase lag days and land memory at different soil depths, as well as the soil memory change with the soil thickness. Furthermore, compared with observations, SSiB3-FSM produces a realistic change of maximum frozen soil depth at decadal scales. This study shows the soil thermal characteristics at seasonal to decadal scales over frozen ground can be greatly improved in SSiB3-FSM and SSiB3-FSM can be used as an effective model for TP and NC simulation during cold reasons.

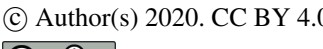
## 1. Introduction

The freeze-thaw process affects the surface thermal characteristics of frozen soil. At short timescales, the freeze-thaw process could delay the winter cooling/spring warming in the frozen soil because of the latent heat received/released through liquid-ice phase change, which affects surface hydrology (Poutou et al., 2004; Li et al., 2010; Bao et al., 2016). At longer timescales, the change of frozen soil and the variations of the freeze-thaw process affect the shrink or expansion of seasonally frozen ground or permafrost, which can affect the active layer or maximum frozen soil depth, water resources (Cuo et al., 2015; Liljedahl et al., 2016; Guo and Wang, 2017) and ecosystems (Yang et al., 2010; Qin et al., 2014; Yi et al., 2014), and is also a crucial response to climate change (Collins et al., 2013; Zhao and Wu, 2019).

Studies have shown that the soil thermal conditions in the frozen ground area of the Tibetan Plateau (TP) and North China (NC) have been experiencing widespread changes since the 1980s, such as a distinct rise in soil temperature at different soil depths (Wu and Zhang, 2008; Zhang et al., 2002) and changes in the soil freeze-thaw processes (Li et al., 2012; Guo and Wang, 2014; Jin et al., 2015; Yang et al., 2018; Li et al., 2020). In recent years, surface water and energy budget modeling on the frozen ground has advanced, especially over the TP (Yang et al., 2009; Zheng et al., 2016, 2017), and current land surface models (LSMs) exhibit improved simulation of soil temperature profiles as soil thaws during the warm monsoon season (Chen et al., 2010; Zeng et al., 2012; Zheng et al., 2014; Cuo et al., 2015). However, more severe warming rates are observed in winter (Zhang et al., 2019), when most LSMs have difficulty in simulating the deep soil temperature and capturing freezing processes over the TP (Su et al., 2013; Zheng et al., 2017). In addition, large discrepancies have been found in the simulation of surface water and energy budgets by different models driven with the same forcing data (Luo et al., 2003; Slater et al., 2007; Zheng et al., 2017) and the most common problem is the systematic under-estimation of soil temperature (Yang et al., 2009; Bi et al., 2016). Unstable simulations are considered to be one of the key obstacles of frozen soil models in frozen ground (Sun, 2005; Bao et al., 2016), and are considered to come from the numerical schemes because the relationship among soil temperature, soil moisture and ice content are highly nonlinear. To date, shortening the time step duration (Flerchinger and Saxton, 1989) and pre-estimating the ice content during numerical iteration are commonly used in frozen soil numerical schemes, but they may make it difficult for the models to reach convergence. Moreover, a heavy computation cost is unavoidable with those approaches. An enthalpy-based soil algorithm was recently applied to solve the nonlinear governing equations in the frozen soil model (Li et al., 2009; Bao et al., 2016). However, it produced a stable solution only at limited sites and has not been tested in regional or global domains.

Moreover, few studies have focused on the model performance based on observed soil temperature anomalies over frozen ground. The ability to preserve of soil temperature anomalies is known as "land memory", which is characterized by exponential decay in amplitude and linear lag in phase of soil temperature with depth. Characteristics of land memory have been documented through observation analysis and modeling studies (Hu and Feng, 2004; Xue et al., 2018; Liu et al., 2020). Hu and Feng (2004) found that the anomaly of soil enthalpy, which can represent integration of soil temperature through the soil column, could persist for 2–3 months in the top 1 m of soil over the eastern United States, and affect





the surface temperature via soil heat flows then affect the variations of summer monsoon rainfall in the
southwest. Another study found the soil enthalpy anomaly in soil column of below 40 cm could persist
for 3–4 months at three sites over the TP (Xue et al., 2018). Over frozen ground, the effects of frozen soil
processes in the land memory are not yet well understood.

Another important issue is the maximum frozen depth (MFD), which occurs during the freezing
period in seasonally frozen ground and can be used to quantify long-term changes in seasonally frozen
ground regions (Zhang et al., 2001). The MFD decreased at 5.58 cm/decade during 1960–2014 over the
TP (Fang et al., 2019). Although the active layer depth (ALD) for permafrost has been investigated over
the TP by different models and compared against field measurements (Oelke and Zhang, 2007; Guo and
Wang, 2013; Li et al., 2020), the MFD has rarely been investigated by models.

Therefore, comprehensive assessments and improvement of the performance of land surface models
for the frozen ground are imperative. In this paper, the third-generation simplified simple biosphere
model (SSiB3) was further improved by coupling with a comprehensive multi-layer frozen soil model
(FSM) (Zhang et al., 2007; Li et al., 2010). By using one host-biophysical model (SSiB3) with freeze-
thaw processes in multi-layer soil (SSiB3-FSM), and comparing its simulated results with observation
data as well as the SSiB3 results, this study focused on the soil temperature profile during freezing and
thawing periods, the change of annual freeze soil depth and its memory capability during the past decades,
to investigate the effects of frozen soil process on these soil thermal characteristics.

This paper is structured as follows. Section 2 describes the models used in this study and the
coupling schemes. Section 3 presents the used data and experimental designs and the calculation methods
of soil temperature memory. The major results obtained in this study, including the characteristics of the
soil temperature profile, variation of MFD and the soil temperature memory, are given in Section 4. A
summary is presented in Section 5.

## 2. Description of the Models and the Coupling Scheme

### 2.1 Background

The SSiB3 model (Xue et al., 1991; Sun and Xue, 2001) substantially enhances the previous model's
ability to simulate cold season temperature dynamics (Xue et al., 2003). However, it only predicts
temperatures of near-surface soil layer ($T_{gs}$) and deep-soil layer ($T_d$) based on the force-restore method
(Deardorff, 1977; Xue et al., 1996). As for the soil water, soil wetness for three soil layers is predicted
and the deepest soil depth is 3.5 m under forests or trees. There are some rough estimations on the soil
freezing and thawing but realistic physical processes in cold season/regions are absent. It is necessary to
introduce a multi-layer frozen soil module based on physical process into SSiB3 for more realistic cold
season/region research under the climate change scenarios.

A comprehensive multi-layer FSM (Zhang et al., 2007; Li et al., 2010), which takes into account of
the interactions between mass and heat transport including ice and liquid water phase exchange, was
coupled with SSiB3 (referred to SSiB3-FSM) for this study.

In the FSM, the freezing-thawing scheme derived from the freezing point depression equation and
the soil matric potential equation is based on thermodynamic equilibrium theory and both liquid water
and ice content have been taken into account in the frozen soil hydrological and thermal property



parameterization. In addition, a variable transformation approach introducing enthalpy and total water

mass in the prognostic equations as substitutes for temperature and liquid water content was used so that

the phase change between liquid and ice can be calculated more efficiently. FSM has previously been

evaluated using observational data from the field station at Rosemount, Minnesota, and many TP sites

with satisfactory results (Li et al., 2009; 2010).

### 2.2. Model coupling scheme

The FSM was implemented into the SSiB3 model to describe multi-layer soil heat transfer and water

flow in SSiB3 affected by freeze-thaw processes in soil. A schematic of the coupled model is shown in

Figure 1. The definition of the symbols in the figure and following equations can be found in Appendix

A. In the soil part, the soil thermal diffusion scheme, soil moisture transport scheme and the freeze-thaw

scheme are designed to solve the soil thermal diffusion, soil water diffusion and ice-liquid phase change,

respectively.

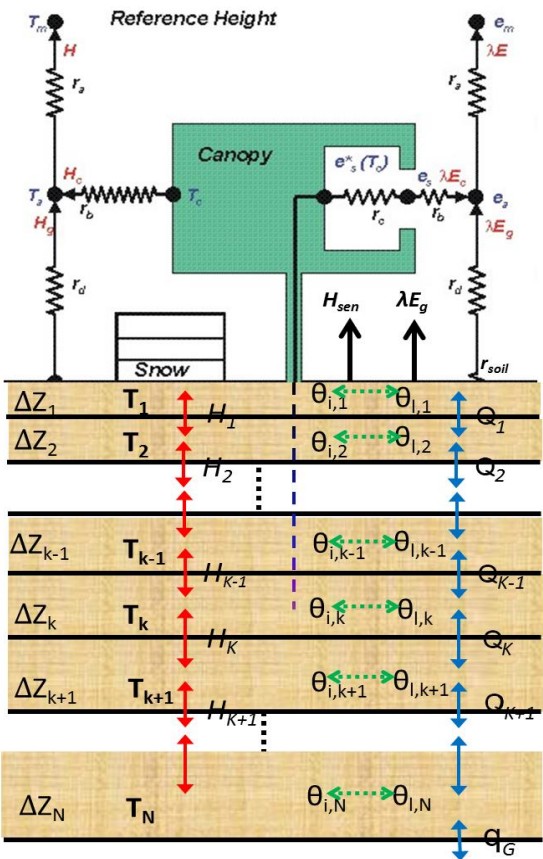


**Fig. 1** Schematic diagram of SSiB3-FSM. Soil temperature, soil volumetric water content and soil

volumetric ice content are $T$, $\theta_l$ and $\theta_i$, respectively. The heat and water flux between soil layers are

represented by $H$ and $Q$. The soil layer number is $k$, which ranges from 1 to $N$.


The soil column is discretized into eight, eleven and twelve layers for desert, grassland and trees, respectively. The thickness of each soil layer increases with the soil depth and the depths of the soil column vary with vegetation types in SSiB3. The surface soil layer was assigned as 2 cm since the variables in the surface are sensitive to the atmospheric diurnal forcing.

### 2.2.1 Energy balance equations

The energy balance equation for canopy indicates that the canopy energy storage change with time is affected by the net radiation at the canopy layer and can be written as:

$$C_c \frac{\partial T_c}{\partial t} = R_{nc} - H_c - \lambda E_c \qquad (1)$$

The heat budget of the uppermost soil layer is affected by the net radiation at soil surface ($R_{ngs}$; W m$^{-2}$), sensible heat ($H_{gs}$; W m$^{-2}$), latent heat fluxes ($\lambda E_{gs}$; W m$^{-2}$), energy exchange with lower soil layer and the phase change between ice and liquid, and can be written as:

$$\frac{\partial(C_s T_{gs})}{\partial t} - L_{il}\rho_i \frac{\partial \theta_i}{\partial t} = \frac{\partial}{\partial z}\left(\lambda_{eff}\frac{\partial T_{gs}}{\partial z}\right) + R_{ngs} - H_{gs} - \lambda E_{gs} \quad (2)$$

The energy distribution inside the soil column is controlled by the heat conduction between layers and the phase change inside each individual layer, so it can be written as:

$$\frac{\partial(C_s T_s)}{\partial t} - L_{il}\rho_i \frac{\partial \theta_i}{\partial t} = \frac{\partial}{\partial z}\left(\lambda_{eff}\frac{\partial T_s}{\partial z}\right) \quad (3)$$

The first term of Eq. (3) on the left is the heat storage change with time in each soil layer. The second term is the latent heat due to freezing/thawing. The first term on the right is the convective heat transferred between the soil layers. At the bottom boundary layer, it is assumed that there is no heat flux from the deeper soil. The differences of energy balance equations for soil between SSiB3 and SSiB3-FSM is the phase change between ice and liquid ($L_{il}\rho_i \frac{\partial \theta_i}{\partial t}$) in the SSiB3-FSM and directly use the heat conduction equation rather than the force-restore method.

### 2.2.2 Water balance equations in soil layers

The water distribution within the soil is driven by the liquid water movement and liquid-ice phase change. This scheme treats the freeze-thaw process as continuous, without a fixed freezing point, and allows the coexistence of water and ice to modify the hydraulic and thermal properties of the soil. The conservation of liquid flow is expressed as a one-dimensional Richards' equation:

$$\frac{\partial \theta_l}{\partial t} = -\frac{\rho_i}{\rho_l}\frac{\partial \theta_i}{\partial t} - \frac{\partial Q_l}{\partial z} - E \quad (4)$$

The liquid water flow rate of $Q_l$ (m s$^{-1}$) is described by Darcy's Law (see (A5) in Appendix B). In the SSiB3-FSM, a freeze-thaw process scheme is used, which is derived from the freezing point depression and soil water potential curve in frozen soil:

$$\theta_l = \theta_s \left[\frac{L_{il}T}{g\psi_0 T_f}(1 + C_k\theta_i)^{-2}\right]^{-\frac{1}{b}} \quad (5)$$

This equation has been employed to describe the relationships among soil temperature, soil liquid water content and ice content (Li et al., 2010).



### 2.2.3 Numerical scheme for the thermal and hydrological equations

Equations (1–5) are highly non-linear systems because the ice content and liquid water change rapidly with little soil temperature change during soil freezing or thawing. We previously substituted soil enthalpy and total water mass for soil temperature and volumetric liquid water content in governing equations (Li et al., 2010) to solve highly nonlinear differential equations. This method also retains energy and water conservation and represents the continuous and slow energy change of the frozen soil system during the freezing/thawing process. However, this approach was only tested for limited field sites. While the method was used in the coupled SSiB3-FSM and tested over a global domain, the numerical solutions become unstable during the long-term integrations for some grid points because the global soil properties and meteorological forcing vary widely. Therefore, a semi-implicit solution procedure for the soil energy and water prognostic equations was developed with SSiB3-FSM for this study.

Figure 2 presents a flow chart of the semi-implicit solution procedure for SSiB3-FSM. A semi-implicit backward finite difference approximation was used for the thermal diffusion equations for canopy and soil (Eq. (1–3)). The numerical Eqs. (A1–A3) are shown in Appendix B. Meanwhile Eq. (5) was transformed to a numerical form (A4) so that it can represent relationship between the change of soil temperature ($\Delta T_s$) and the change of soil ice content ($\Delta\theta_i$) assuming the total water mass is conserved during one time step. Then a tridiagonal linear equation system (A5) for the change of soil temperature was derived based on Eqs. (A1–A3, A4). After solving the tridiagonal matrix at different soil layers, the phase change between liquid water and ice ($\Delta\theta_i$) in soil was decided using the change of soil temperature ($\Delta T_s$) during one time step (Eq. A4). Because the phase change has been included while solving the temperature tridiagonal matrix, here obtained the soil water content ($\theta_i$) and soil temperature ($T_s$) for each soil layer. After the change of ice content has been decided, the water balance equations do not involve the prognostic variable of the ice content. Subsequently, we can solve the tridiagonal matrix for water fluxes at the interface of the soil layers (see Appendix B (A6–A10)). The liquid water content at the current time step can be easily obtained from Eq. (A11).

This semi-implicit scheme for soil temperature, liquid water and ice content in SSiB3-FSM has been tested over the global domain. It can effectively produce stable solutions for long-term integrations with the heat and mass balances.





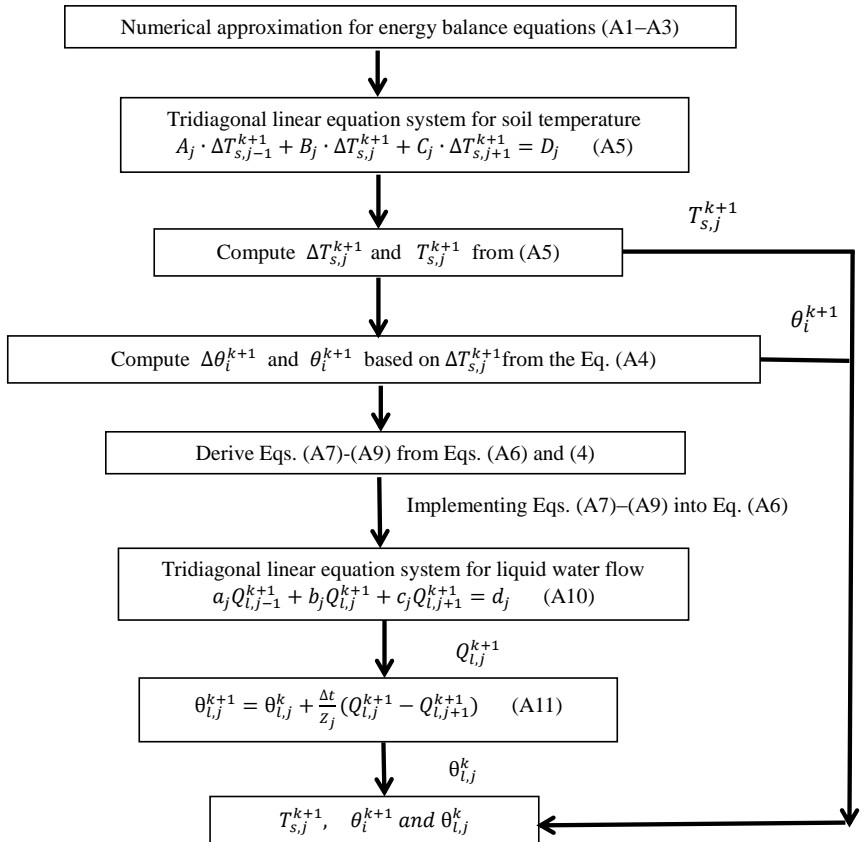

**Fig. 2** Flowchart of the semi-implicit solution procedure for SSiB3-FSM.

## 3. Data Sets and Experimental Design

### 3.1 Data sets

From 1948 to 2007, the SSiB3 model and coupled offline SSiB3-FSM model have been driven using the meteorological forcing from the Princeton global meteorological data set (Sheffield et al., 2006), which is developed by combining a suite of global observation-based data sets with the National Centers for Environmental Prediction/National Center for Atmospheric Research reanalysis data. The data set includes surface air temperature, pressure, specific humidity, wind speed, downward short-wave

radiation flux, downward long-wave radiation flux and precipitation. The spatial resolution is $1° \times 1°$, and the temporal resolution is 3 h.

Several observation data sets have been used to evaluate the performance of SSiB3-FSM and SSiB3 in cold regions. For the near surface temperature ($T_{2m}$), we used the Global Historical Climatology Network version 2 and the Climate Anomaly Monitoring System (GHCN-CAMS) gauge-based 2-m

temperature over land for 1979–2007, which provides global coverage of monthly means on a regular resolution of 0.5° latitude × 0.5° longitude grids (Fan and van den Dool, 2008).

For the soil temperature profile and the MFD over the TP and NC, the monthly mean soil temperature

of 626 stations over China for 1981–2005 has been used (Yang and Zhang, 2016), provided by the China

Meteorological Administration. The data set has nine soil layers at 0, 5, 10, 15, 20, 40, 80, 160 and 320

cm.

Since the calculation of land soil temperature memory requires a long time series of soil temperature

data, only the stations with complete records for 1981–2005 and nine soil layers over the TP region

(elevation > 2500 m) and NC (110°E–120°E, 34.5°N–42°N) were selected. There are 14 stations over

the TP and 16 sites over NC used for this study. Fig. 3 shows the spatial distribution of the stations with

available soil temperature data for all nine soil layers and for all 12 months of all 25 years over the TP

and NC.

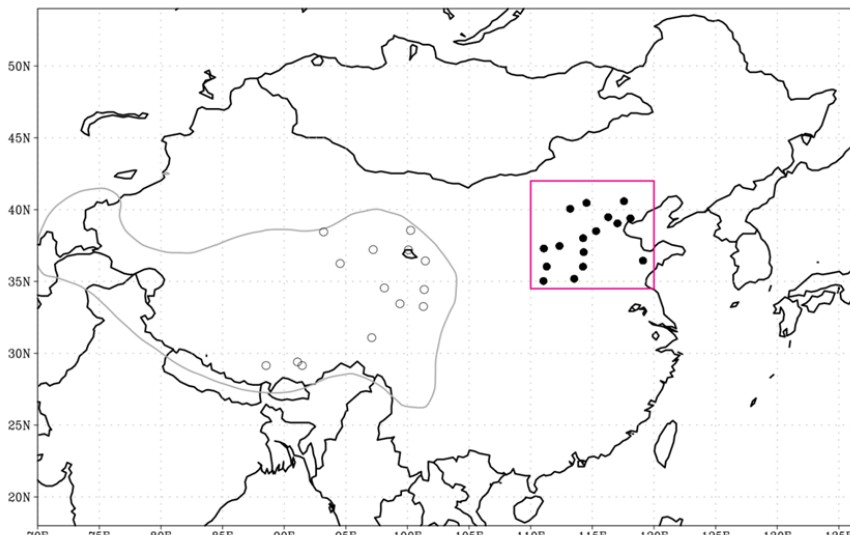

**Fig. 3** Geographical distribution of stations with complete soil temperature records for all nine soil

layers for 1981–2005. The red boxed region is North China (110°E–120°E, 34.5°N–42°N) and the

locations of 16 sites used in this study are marked by solid circles. The grey line represents the elevation

is above 2500 m and the grey empty circles denote the locations of the 14 sites on the TP.

### 3.2 Experimental Design and Methods

#### 3.2.1 Control run

A global simulation by SSiB3-FSM and SSiB3 was carried out forced by the Princeton global

meteorological data set from 1948–2007. The initial soil temperature and liquid water content profiles

were derived by interpolating the NCEP-DOE Reanalysis II (R2) (NCEP-R2, Kanamitsu et al., 2002)

soil temperature and soil moisture data linearly to the model's soil layers. Because the soil ice content

measurements are unavailable, and the initial soil ice content is essential for the soil hydrological and

thermal properties, we set ice content to zero at the beginning. The first 10 years (1948–1957) were used

for model spin up and the simulation for the last 50 years (1958–2007) was analyzed. The observational





data were used to evaluate the model performance and the results from this run were used to analyze the cold regions' thermal characteristics and MFD, as well as their variations under global warming.

### 3.2.2 Sensitivity run

To investigate the sensitivity of the soil temperature profile and other thermal characteristics to the freeze-thaw process, we conducted a sensitive simulation using the SSiB3-FSM under the same initial land surface conditions but without freeze-thaw process in soil. This sensitivity run is referred to as SSiB3-FSMnoICE run hereafter. Both the SSiB3-FSM run and the SSiB3-FSMnoICE run produce multi-layer's soil temperature and soil moisture, MFD, net radiation, latent heat flux and sensible heat flux, as well as the canopy temperature, canopy water and interception.

### 3.2.3 Methodology to determine MFD and soil memory

Based on the classification of the permafrost by Frauenfeld et al. (2004), a site was deemed to be seasonally frozen ground while the soil temperature at 3.2 m is above 0°C. Based on this criterion, the 14 stations over the TP and the 16 sites over NC in this study were all classified as seasonally frozen ground. According to the seasonal characteristic of soil temperature over seasonally frozen ground, the 240   MFD for each year can be defined as an index for the study of seasonal frozen soil variability and change. This paper gives a preliminary estimation of MFD variations based on monthly soil temperature. Following Frauenfeld et al. (2004), the maximum depth of zero isothermal line for some year is defined as the MFD for this year. Frauenfeld et al. (2004) validated this robustness of this approach. It should be noted that the MFD is different from active layer thickness (ALT), because the active layer is defined as 245   "the top layer of ground subject to annual thawing and freezing in areas underlain by permafrost" (van Everdingen, 1998). ALT is suitable for the permafrost but MFD is more suitable for the seasonally frozen ground. As the ALT increases, the permafrost thaws deeper, whereas as the MFD increases, the frozen soil freezes deeper.

The persistence of soil temperature can be quantified by temporal scale analysis. Hu and Feng (2004) 250   assumed that the temporal variation of the soil enthalpy in North America followed the first-order Markov process. Instead of analyzing soil temperature only, the variations of soil enthalpy, which represents integration of soil temperature through the soil column, was used to examine the land memory (Hu and Feng, 2004). This study uses the observed and simulated soil temperature from ground stations in NC and the TP, and the method presented in Entin et al. (2000) and Hu and Feng (2004) to analyze the 255   persistence. Land memory is characterized by the variable's autocorrelation, $r$, satisfying the following:

$$r(\delta t) = \exp(\frac{-\delta t}{S}) \qquad (6)$$

in which $\delta t$ is the time lag, $S$ is the decay time scale that can characterize a red noise process and $r(\delta t)$ is the autocorrelation coefficient at the lag time (e.g., 1 month, 2 months, 3 months, …)

### 4. Results and Discussions

### 4.1 Assessment of simulated surface 2 m temperature and temperature profile

#### 4.1.1 Surface 2 m temperature





Before investigating the frozen soil thermal characteristics and MFD, as well as their variability, the SSiB-FSM is first evaluated using the observational data. The root mean square error (RMSE) and absolute bias (BIAS) between CAMS and simulated surface temperature globally as well as TP and NC

from the SSiB3-FSM and SSiB3 are assessed (Table 1). Table 1 shows the annual RMSE and BIAS of SSiB3-FSM are less than those of SSiB3. In addition, in different seasons the SSiB3-FSM shows less bias than SSiB3. Overall, the SSiB3-FSM produces more realistic estimates of surface temperature than SSiB3 and it can predict the heat transfer processes globally and locally with a reliable accuracy, which provides a basis for further discussions.

**Table 1. Error statistics of the simulated surface temperature by SSiB3-FSM and SSiB3.**

| Global | BIAS (°C) | | RMSE (°C) | |
|---|---|---|---|---|
| | SSiB3 | SSiB3-FSM | SSiB3 | SSiB3-FSM |
| Annual | 1.27 | 0.97 | 2.25 | 1.93 |
| MAM | 1.79 | 1.66 | 2.71 | 2.47 |
| JJA | 1.81 | 1.37 | 2.87 | 2.42 |
| SON | 0.80 | 0.29 | 2.34 | 2.16 |
| DJF | 0.64 | 0.51 | 2.47 | 2.24 |

**(2) Soil temperature profile in the TP**

The seasonal vertical soil temperature profile strongly mirrors the influence of the air temperature forcing in contrast to the almost isothermal average annual temperature profile (Oelke and Zhang, 2004). The averaged seasonal profiles for observed soil temperature at 14 sites over the TP are shown in Fig. 4a.

For easy comparison, only soil temperature profiles in Jan., Apr., July and Oct. are shown, since the four curves represent the characteristics of soil temperature profile in winter, spring, summer and autumn, respectively. At the seasonal scale, the surface layer and the subsurface layers (here referred to surface to ~1 m) are frozen during winter (Jan.), whereas the temperature of deep layers (below 1 m) is above 0°C. The surface and subsurface soil begins to thaw from Mar.. In Apr. the soil is almost unfrozen. Deep

soil temperature always stays above 0°C.

There is a generally rising trend in monthly temperature from winter to summer (from Jan. to July) above 2 m soil depth. Between the 2 m and 2.5 m depth there are no apparent changes in soil temperature from Jan. to Apr. Below 2.5 m, there is an inverse trend compared with upper soil temperature. For instance, the temperature in Jan. is higher than that in Apr. at 3 m soil depth. From Apr. to July, the

temperature of all soil layers (surface to ~3.2 m) increases with the air temperature due to the increasing solar radiation. As the autumn approaches, the soil temperature above 1.5 m begins to decrease. Below



1.5 m there is a time lag behind the rising trend, leading to higher temperature at 3.2 m in Oct. than that
in July. From Oct. to Jan., soil temperature on all layers shows a decreasing trend. Generally, the soil is
characterized as seasonally frozen ground. Deep soil temperature shows a time lag compared with the
surface layer, and the upper soil temperature (<1.5 m) shows larger seasonal variability than the deep soil
temperature.

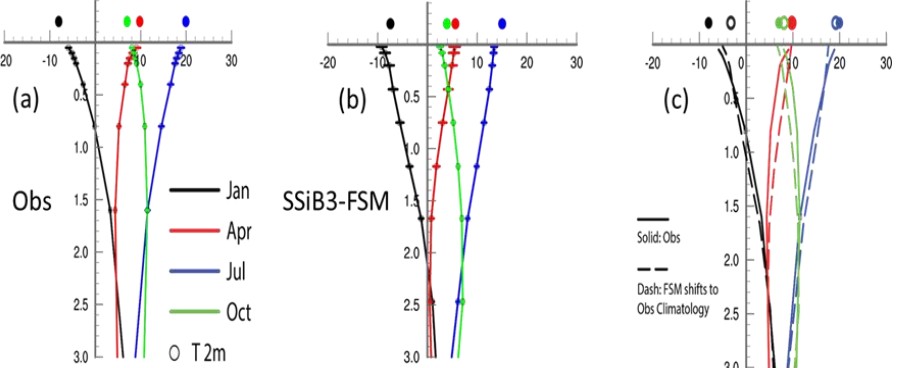

**Fig. 4** The seasonal soil temperature profile over the TP (14 sites) for 1981–2005. (a) observation;
(b) simulated by SSiB3-FSM; (c) comparison between the observation and the SSiB3-FSM shifts to the
observation climatology.

The simulated soil temperature profile by SSiB3-FSM over the TP in different seasons is shown in
Fig. 4b. There is a general consistency between the simulated temperature profile and the observed profile
in both vertical distribution and the seasonal variations. Compared against observations, however, the
simulated soil temperatures underestimate the temperature in whole soil column throughout all seasons.
The air temperature at 2 m in Apr., July and Oct. in SSiB3-FSM is lower than that observed. This
systematic bias arises from the forcing data. For example, the observed air temperature in Apr. at 2 m is
about 10°C but the forcing for the models is only 6°C. The greatest difference (about 6°C) is in July.
Considering the forcing data's bias, we can parallelly move the SSiB3-FSM soil temperature profile to
make the simulated and the observed 2m temperature climatology at the same position. Subsequently,
the observed and simulated soil temperature profiles almost coincided (Fig. 4c).

**(3) Soil temperature profile in NC**

The simulated and observed seasonal soil temperature profiles over NC are displayed in Fig. 5a
and Fig. 5b, respectively. They show a similar seasonal frozen soil temperature variability to those over
the TP but its thawing season is earlier than that of TP. As the observations show, in winter (Jan.), the
soil freezes above 40 cm and it begins to thaw in Feb. until Mar. The soil under 40 cm stays above 0°C
throughout the year. The simulated temperature profiles and their seasonal variations are adequately
consistent with the observations. However, the frozen depth is deeper in SSiB3-FSM than that of the
observations in Jan. (Fig. 5c). These differences may be attributed to the parameterization of soil thermal
and hydrological process in SSiB3-FSM.



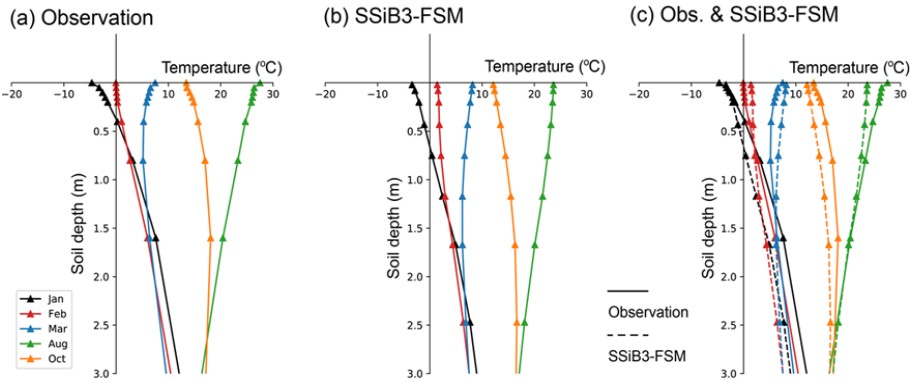


**Fig. 5** The seasonal soil temperature profile over NC (16 sites) for 1981–2005. (a) Observation;
(b) simulated by SSiB3-FSM; (c) observation and SSiB3-FSM.

**(4) Comparison with the force-restore method**

In SSiB3 with the force-restore method, only surface temperature and deep soil temperature are

considered. For the seasonal change of soil temperature, both the seasonal variation of surface soil temperature ($T_{gs}$) and deep soil temperature ($T_d$) (Fig. 6a and 6c) are on the same phase, only with a weak lag in $T_d$. It is difficult to define its precise position of the deep soil temperature layer, which is dependent on the vegetation types and soil conditions. By introducing a multi-layer frozen soil model, SSiB3-FSM not only presents more precise soil temperature profile but also clearly shows the seasonal change of soil

temperature at different soil depths (Fig. 6b and 6d). The time lag and amplitude damping of temperature become more pronounced with increasing depth and they are well described in SSiB3-FSM (Fig. 6b and 6d). This improved performance of SSiB3-FSM lays a foundation for further investigation of the characteristics of MFD changes and soil memory.



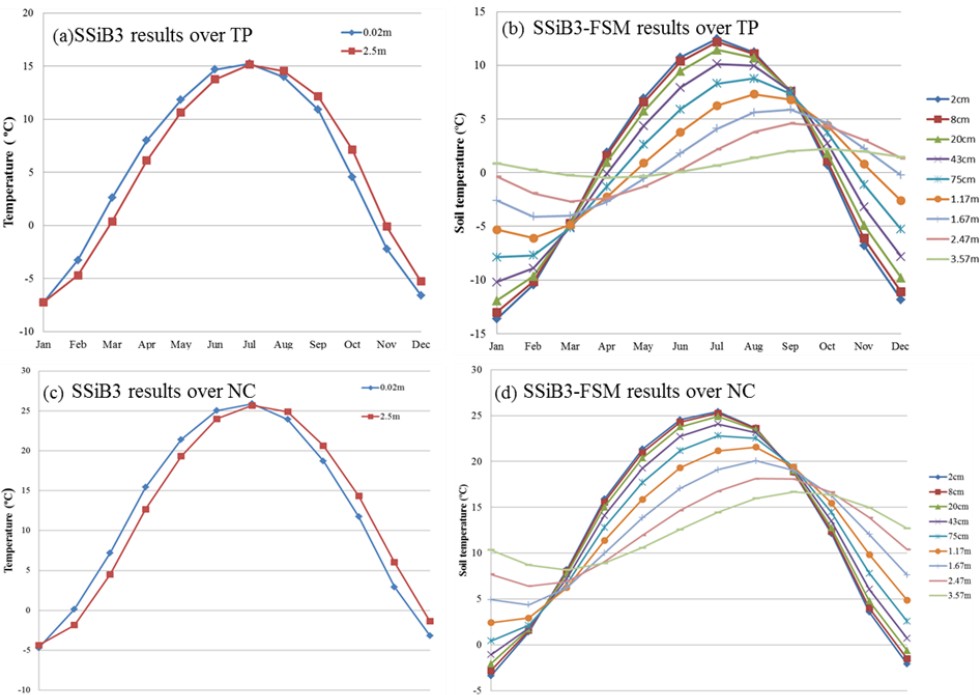

**Fig. 6** The seasonal soil temperature simulated by SSiB3 and SSiB-FSM over the TP (14 sites) and NC (16 sites) for 1981–2005. (a) The seasonal climatology of $T_{gs}$ (0.02 m) and $T_d$ (2.5 m) by SSiB3 over the TP; (b) the seasonal temperature climatology by SSiB3-FSM over the TP; (c) the seasonal climatology of $T_{gs}$ (0.02 m) and $T_d$ (2.5 m) by SSiB3 over NC, and (d) the seasonal temperature climatology by SSiB3-FSM over NC.

### 4.2. Characteristics of the soil temperature profile over the TP and NC

#### 4.2.1 Temporal variability of the soil temperature profile over the TP and NC

The temporal variability with depth of soil temperature was further explored by analyzing the phase variations with depth. Here, we used the cross-correlation statistical method to analyze how the seasonal variability in soil temperature decreases with depth. Because the SSiB3-FSM produced reasonable surface and subsurface temperature profiles (as discussed in Section 4.1) and the observational data are only at monthly resolution, the 50-year simulated daily soil temperature were used to represent the time-space variability of soil temperature over a wide area. Fig. 7a and Fig. 8a show the lag cross-correlations between soil temperature of the first layer with other layers over the TP and NC, respectively. The time lag at which the maximal correlations occur increases with soil depth. For instance, the soil layer at 59 cm reaches maximum at about 10 days while the layer at 312 cm needs about 90 days (a season). The cross-correlation values decrease after they reach the maximum value and reach zero at about 110 and 180 days for the soil layer at 59 and 312 cm, respectively. To more clearly display these relationships, the soil temperature phase lag time, defined as the point at which the cross-correlation with the first soil





layer equals 1, with depth is shown in Fig. 7b and Fig. 8b. The phase lag time increases linearly with the

soil depth, the details of which are presented in Table 2. For the soil depth at 1.5 m over the TP, the phase

lag for soil temperature is about 43 days (~1.5 months). For the soil depth at 3 m, the phase lag could be

87 days (~3 months). Over NC, the phase lag for soil temperature is 11 and 32 days at 59 cm and 1.5 m,

respectively.

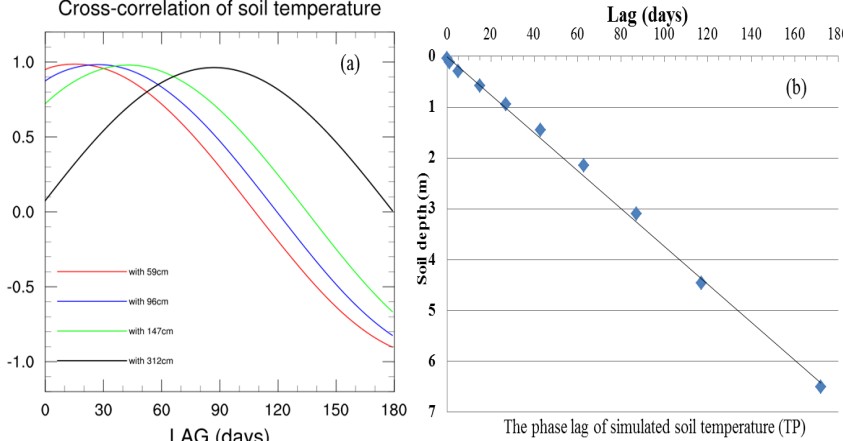

**Fig. 7** The time–space variability of soil temperature over the TP for 1978–2007. (a) Simulated

cross-correlation of first layer soil temperature with other soil layers (red line: 59 cm; purple line: 96 cm;

green line: 147 cm; black line: 312 cm) by SSiB3-FSM; (b) the phase lag (days) of simulated temperature

over the TP by SSiB3-FSM.

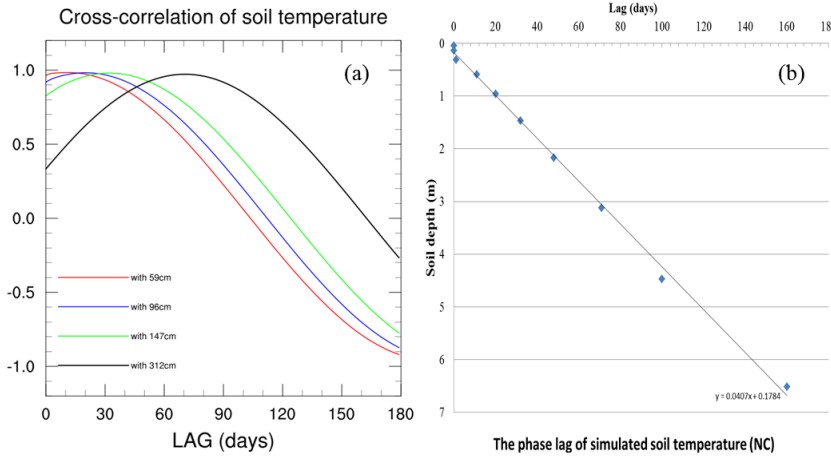

**Fig. 8** The time–space variability of soil temperature over NC for 1978–2007. (a) Simulated cross-

correlation of the first layer soil temperature with other soil layers (red line: 59 cm; purple line: 96 cm;

green line: 147 cm; black line: 312 cm) by SSiB3-FSM; (b) the phase lag (days) of simulated temperature

over NC by SSiB3-FSM.






**Table 2. Phase lag (days) of simulated soil temperature at different soil depths by SSiB3-FSM and SSiB-FSMnoICE.**

| Depth (m) | Phase lag (days) | | | |
| | TP | | NC | |
| | SSiB3-FSM | SSiB3-FSMnoICE | SSiB3-FSM | SSiB3-FSMnoICE |
|---|---|---|---|---|
| 0.59 | 15 | 12 | 11 | 9 |
| 0.96 | 27 | 22 | 20 | 19 |
| 1.47 | 43 | 35 | 32 | 31 |
| 3.12 | 87 | 72 | 71 | 71 |
| 4.47 | 117 | 103 | 100 | 100 |

### 4.2.2 Land memory

The land surface temperature anomaly over the TP and North America has been recognized as an
indicator for extreme hydroclimate events (Xue et al., 2018) because of its preservation of the snow and
other climate signatures in previous months. Evaluating the soil persistence of SSiB3 and SSiB3-FSM
and comparison with observed soil memory are crucial for its application in climate studies. The above
analyses of temporal variability in soil temperature with depth shows the soil temperature simulated by
SSiB3-FSM is characterized by increasing persistence with depth. This suggests SSiB3-FSM can be used
to study the land persistence of soil enthalpy, which represents integration of soil temperature through
the soil column. The land persistence of soil enthalpy over the TP has been preliminary investigated by
Xue et al. (2018) and another paper (Liu et al., 2020). The current paper therefore focuses on analyzing
the land memory over NC.

Taking the natural log on both sides of Eq. (6) and rearranging, we can obtain $\delta t = -S\ln[r(\delta t)]$, which
describes a straight line in the two-dimensional domain of $\delta t$ and the natural log of autocorrelation, $r$.
Following this procedure, we calculated autocorrelations of observed monthly soil enthalpy anomalies
between 5 and 320 cm at time lags from 1 to 4 months at the 16 stations over NC and plotted their average
autocorrelations in the $\delta t$-$\ln[r(\delta t)]$ domain (Fig. 9a). The lagged autocorrelations of the simulated
monthly soil enthalpy anomalies at different soil layers between 5 and 312 cm were also calculated and
are shown in Fig. 9b. The persistence of soil enthalpy anomalies is determined by the negative inverse
of the slope of the straight line for each case in Fig. 9. The slope of these lines varies, indicating a different
persistence time of the soil enthalpy anomaly at different depths.



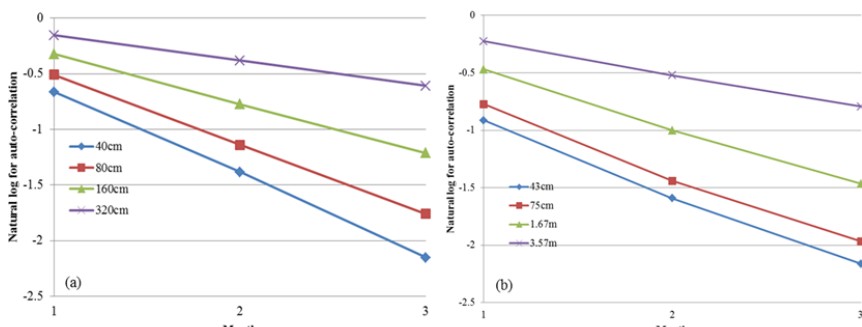

**Fig. 9** The natural log for the auto-correlation of over NC for 1981–2005. (a) Observed; (b) SSiB3-
FSM.

The persistence values over NC are given in Table 3. For the observations, the persistence of soil
enthalpy anomalies is about 1.34 months in the top 40 cm column and increases to longer than 2 months
in the top 160 cm column under the soil surface. In the 320 cm soil column the persistence of soil enthalpy
anomalies reaches 4.4 months. Similarly, the persistence of simulated soil enthalpy anomalies is 1.37
months in the 43 cm below the surface. It increases to longer than 2 months about 167 cm soil column.
The observed persistence change with the soil thickness is reasonably simulated with the SSiB3-FSM.

**Table 3. The persistence of soil temperature at different soil depths by SSiB3-FSM and the
observations.**

| Depth(m) | Persistence (month) | |
|---|---|---|
| (obs./model) | Obs. | SSiB3-FSM |
| 0.40/0.43 | 1.34 | 1.37 |
| 0.80/0.75 | 1.60 | 1.50 |
| 1.60/1.67 | 2.25 | 2.13 |
| 3.20/3.5 | 4.4 | 4.75 |

The persistence by SSiB3 with the force-restore method is only 1.2 and 1.23 months for the surface
temperature and deep temperature, respectively. SSiB3-FSM shows better performance in simulating the
land persistence for the anomalies in soil than SSiB3, which only considers two layers of temperature
data.

**4.3 Sensitivity of the soil temperature profile to the freeze-thaw process**

A sensitivity experiment, test-SSiB3-FSMnoICE, in which the freeze-thaw process in soil is not
included, was detailed in Section 3.2.2. A comparison of soil temperature profiles was made between
SSiB3-FSM and SSiB3-FSMnoICE and the results are shown in Fig. 10. With freeze-thaw
parameterization, the latent heat released while freezing, e.g., in Oct., could offset the decreasing soil
temperature. The soil temperature, therefore, would be higher than that of SSiB3-FSMnoICE. Over the
TP, the largest difference is found for Jan., especially between 50 cm and 1.5 m. The simulated soil
temperature by SSiB3-FSM is about 1–1.7°C higher than that by SSiB3-FSMnoICE. Over NC, the large



differences are also shown in winter, especially in Jan. The difference in Jan. is about 1−1.2°C from 1.5 cm to 1.5 m, which means the freeze process in soil delays the winter cooling in freezing seasons (from Oct. to Jan.), and delays the summer warming in thawing seasons (from Apr. to July).

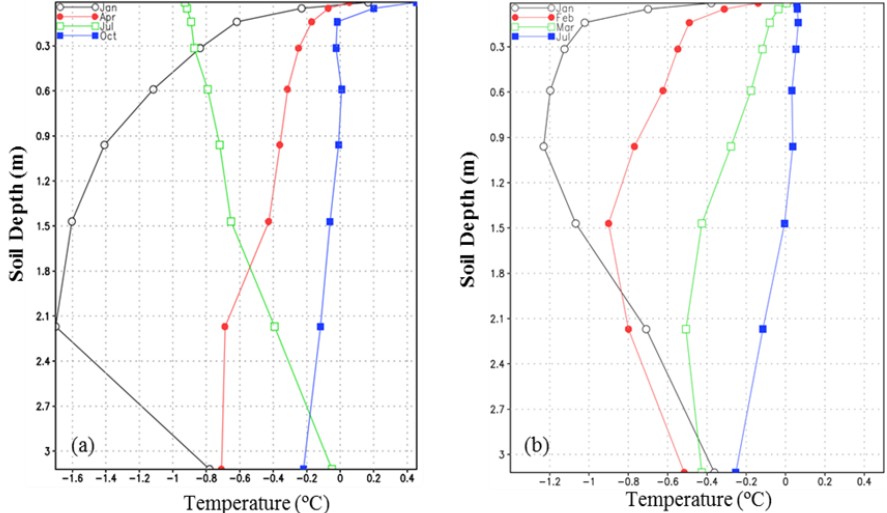


**Fig. 10** Differences in the seasonal soil temperature profile between the SSiB3-FSMnoICE and SSiB3-FSM for 1981–2005 over (a) the TP (14 sites) and (b) NC (16 sites).

The freeze-thaw process has a major impact on soil temperature profile simulation, especially when freezing or thawing occurs. This effect would exert an impact on the spatial and temporal variability of soil temperature and play an essential role in the soil temperature time lag. Table 2 shows that the time lag of SSiB3-FSMnoICE is less than that of SSiB3-FSM in almost every soil layer. In particular, over the TP, the difference of lag days between the SSiB3-FSM scheme and SSiB3-FSMnoICE scheme increases with depth. For instance, at 59 cm, the time lag of SSiB3-FSM scheme is 3 days longer than that of no ice scheme; however, at about 3 m depth, the difference is about 15 days. For NC, the freeze-thaw processes also increase the phase lag days even though the number of phase lag days between the SSiB3-FSM and SSiB3-FSMnoICE show less difference than those over the TP in the upper soil depths. This is because the maximum freezing depth over NC is about 30 cm, much shallower than that over the TP. Correspondingly, the effects of the freeze-thaw process are only exerted at shallower soil depths.

### 4.4 MFD over the TP

The long-term temperature profiles at 14 stations over the TP and 16 sites over NC exhibit characteristics of seasonally frozen ground, which freezes in winter and thaws in spring at the surface soil and remains unfrozen at 3.2 m depth throughout the entire year. The simulated annual soil temporal variation with soil depth over the TP and NC stations are shown in Fig. 11, which displays the seasonal freezing and thawing processes during 1981–2005 for the TP and NC. Over the TP, the surface soil starts to freeze in the middle of Oct. and the MFD occurs around Apr. at 1.8 m. For NC, the surface soil starts



to freeze at the beginning of Dec. and the MFD occurs in Feb. at around 60 cm soil depth, which is much shallower than that of the TP.

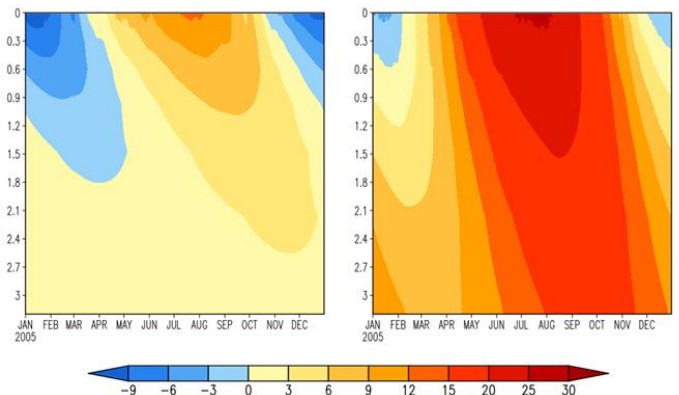

**Fig. 11** The climatology of simulated daily soil temperature (Left) over the TP (14 sites averaged)
and (Right) NC (16 sites averaged) for 1981–2005 (Unit: °C).

The MFD at 14 sites over the TP and 16 sites over NC simulated by SSiB3-FSM was averaged to analyze the changes of the MFD for 1981-2005. Zero-score normalization, which is a commonly used normalization method, was employed to normalize the observed and simulated MFD:

$$y_i = \frac{x_i - \bar{x}}{s} \quad (7)$$


$$\bar{x} = \frac{1}{n}\sum_{i=1}^{n} x_i \, , s = \sqrt{\frac{1}{n-1}\sum_{i=1}^{n}(x_i - \bar{x})^2} \quad (8)$$

where $x_i$ is the original MFD value for 1981–2005 and $y_i$ is the normalized MFD corresponding to $x_i$.

Both the observed MFD over the TP and NC showed a significant decreasing trend from 1981 to 2005 (Fig. 12). These decreasing trends indicate that, in areas of seasonally frozen ground, the freezing ground became increasingly shallower during this time period. Over the TP, the observed net change is
a 23 cm decrease in MFD in 2005 compared with 1981 and the rate of decrease is about 0.92 cm year$^{-1}$. However, from 1983 to 1990 the rate is about 4.5 cm year$^{-1}$, which is about four times as much as that in 1981–2005. In the 1990s, the decreasing rate is about 3 cm year$^{-1}$. After the 2000s, the decreasing trend reduced. A similar decreasing in the simulated normalized deviation of MFD at TP 14 sites for 1981–2005 is shown in Fig. 12a. The rate of decrease intensified during 1983–1990, which was also
shown in the observations.

Over NC, the observed decrease in MFD is 13 cm from 1981 to 2005. The highest decreasing rate is about 3.1 cm year$^{-1}$ from 1981 to 1990, about six times higher than in 1981–2005. The simulated results by SSiB3-FSM also show the consistency with the observations, especially during the 1980s, when the MFD decreasing trend is 1.8 cm year$^{-1}$.
The MFD decreasing trend during the 1980s over the TP may be related to the significant increasing trend in the winter and spring air temperature. Wei et al. (2003) analyzed the inter-decadal



variations of air temperature over the TP and found a global climatic jump in the 1980s on the TP and
the air temperature increased more strongly in the winter and spring from the 1980s to 2000s. They
showed that the rates of increasing air temperature at most stations were 0.02–0.04°C year$^{-1}$ in the winter

and spring, which leads to an increasing trend in the 10–20 cm soil temperature over the TP (Zhang et
al., 2008). For the NC, an increasing winter temperature trend has been detected since 1985 (Zhang et
al., 2002; Shen et al., 2010), which may lead to the decreasing MFD over NC since 1980s.

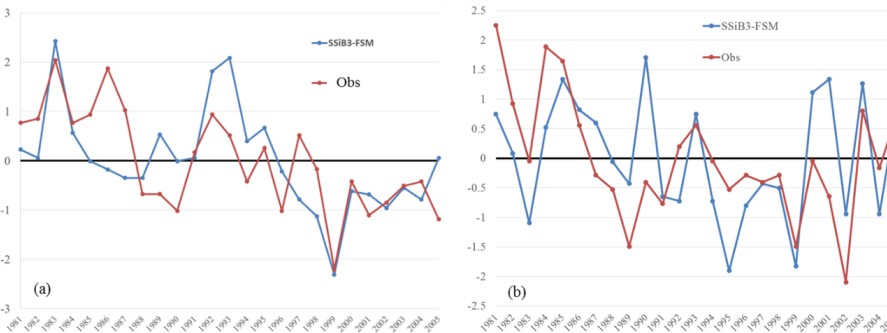

**Fig. 12** Normalization of MFD over (a) the TP and (b) NC for 1981–2005 for SSiB3-FSM and the

observations.

A comparison of MFD between SSiB3-FSM and SSiB3-FSMnoICE was conducted to evaluate the
effects of freeze-thaw processes on the MFD. Although the heat and water mass balance due to ice–liquid
phase change is not included in SSiB3-FSMnoICE, the soil temperature still experience a large range of
variation. Both MFDs show almost the same variations but the MFD in SSiB3-FSM is shallower than

that in SSiB3-FSMnoICE (figure not presented). This can be explained by the phase change energy
released while freezing, which could offset the decreasing temperature during the freezing period and
lead to a higher soil temperature at the same soil depth than that simulated by the SSiB3-FSMnoICE.

**5. Conclusions**

To improve the accuracy of soil temperature simulation in frozen ground, a multi-layer FSM was

incorporated to the SSiB3 to represent the freezing-thawing process and the heat and water transfer in a
multi-layer frozen soil. By introducing a semi-implicit backward finite difference approximation and a
freezing-thawing scheme based on the freezing depression equation, the highly nonlinear equations in
multi-layer frozen soil can be efficiently and stably solved by two tridiagonal matrixes in SSiB3-FSM.
The simulated results show that with the frozen soil component, the SSiB3-FSM produces realistic soil

thermal characteristics than that of SSiB3, especially soil vertical temperature profiles in different
seasons.

Furthermore, our results confirm the important role of frozen soil processes on soil thermal
characteristics at different time scales over NC and the TP. The results show that the phase-change latent
heat released while freezing can offset the decreasing soil temperature, therefore the soil temperature

could be higher than that of the experiment without the frozen soil process as soil freezes. Further analysis
into the spatial and temporal variability of soil temperature showed that the seasonal variability of soil



temperature decreases with soil depths and the phase lag damps linearly. The frozen soil process could increase the phase lag of soil temperature from several days in the surface layer to 15 days in deep layers.

The investigation of SSiB3-FSM's ability to simulate the variability of maximum frozen depth at decadal scales showed that simulated normalized of MFD over the TP and NC by SSiB3-FSM are in good agreement with the observations for 1981–2005, including the substantial decreasing trends and the variabilities at decadal scales. The frozen soil processes affect the magnitudes but do not change the decreasing trends of MFD. SSiB3-FSM shows shallower MFD than SSiB3-FSMnoICE because the simulated soil temperature in SSiB3-FSM is higher than that in SSiB3-FSMnoICE. In addition, the

SSiB3-FSM also can reproduce the reliable soil memory at different soil depths.

The changes of soil properties and their parameterizations have great effects on the surface energy balance. In particular, the soil thermal conductivity shows large spatial variabilities, and the soil thermal properties are heterogeneous in the vertical direction. A disparity of the soil properties between models and observations may result in the difference between the observations and the simulations.

Although the SSiB3-FSM is capable of capturing the basic soil thermal characteristics at seasonal and decadal scales over regions of seasonally frozen ground, further analyses into soil hydrological characteristics in the freezing and thawing phases remain to be conducted. In addition, the better performance of SSiB3-FSM than that by the SSiB3 or SSiB3-FSMnoICE is not only attributed to the frozen soil process but also to the multi-layer heat and water transfer scheme. The effects of the soil

stratification and the soil column depth on the model's performance over seasonal frozen ground require further study.






**Appendix A. List of symbols with units and definition**

| Symbol | Units | Definition |
|--------|-------|------------|
| $b$ | – | Exponent in the Clapp and Hornberger (1978) parameterization |
| $C_c$ | J m$^{-3}$ K$^{-1}$ | Canopy volumetric heat capacity |
| $C_k$ | – | An adjustable constant parameter in Eq. (5) |
| $C_s$ | J m$^{-3}$ K$^{-1}$ | Soil volumetric heat capacity |
| $E$ | m$^3$ m$^{-3}$ s$^{-1}$ | Evaporation for the surface soil or transpiration for the soil root zone layers |
| $E_{ct}$ | m$^3$ m$^{-3}$ s$^{-1}$ | Canopy transpiration |
| $E_{gs}$ | m$^3$ m$^{-3}$ s$^{-1}$ | Soil surface evaporation |
| $\lambda E_c$ | W m$^{-2}$ | Latent heat flux at the canopy layer |
| $\lambda E_{gs}$ | W m$^{-2}$ | Latent heat flux at the soil surface |
| $g$ | m s$^{-2}$ | Gravity |
| $H_c$ | W m$^{-2}$ | Sensible heat flux at the canopy layer |
| $H_{gs}$ | W m$^{-2}$ | Sensible heat flux at the soil surface |
| $L_{il}$ | J kg$^{-1}$ | Specific latent heat of fusion |
| $Q_l$ | m s$^{-1}$ | Liquid water flow rate |
| $q_G$ | m s$^{-1}$ | Gravitational drainage at the bottom soil layer |
| $R_{nc}$ | W m$^{-2}$ | Net solar radiation flux at the canopy layer |
| $R_{ngs}$ | W m$^{-2}$ | Net solar radiation flux at the soil surface |
| $t$ | s | Time |
| $T_c$ | K | Canopy temperature |
| $T_d$ | K | Deep-soil layer temperature in SSiB3 |
| $T_f$ | K | Freezing temperature (273.15) |
| $T_{gs}$ | K | Near-surface soil layer temperature in SSiB3 |
| $T_s$ | K | Soil temperature for inner soil layers |
| $\rho_l$ | kg m$^{-3}$ | Density of liquid water |
| $\rho_i$ | kg m$^{-3}$ | Density of ice |
| $\theta_l$ | m$^3$ m$^{-3}$ | Soil volumetric liquid water content |
| $\theta_i$ | m$^3$ m$^{-3}$ | Soil volumetric ice content |
| $\theta_s$ | m$^3$ m$^{-3}$ | Soil porosity |
| $\theta_T$ | m$^3$ m$^{-3}$ | Total soil water during one time step |
| $\lambda_{eff}$ | W m$^{-1}$ K$^{-1}$ | Effective soil thermal conductivity |
| $\psi_0$ | m | Soil saturated water potential |
| $\psi$ | m | Soil matric potential |

**Appendix B. Numerical scheme for solving governing equations in SSiB3-FSM**

In SSiB3-FSM, a semi-implicit backward finite difference approximation was used for the thermal diffusion in the soil.

For energy balance equation at canopy layer, Eq. (1) can be written as:


$$C_c \frac{\Delta T_c}{\Delta t} = R_{nc} + \frac{\partial R_{nc}}{\partial T_c} \cdot \Delta T_c + \frac{\partial R_{nc}}{\partial T_{gs}} \cdot \Delta T_{gs}$$

$$- H_c - \frac{\partial H_c}{\partial T_c} \cdot \Delta T_c - \frac{\partial H_c}{\partial T_{gs}} \cdot \Delta T_{gs} \qquad \text{(A1)}$$

$$- \lambda E_c - \lambda \frac{\partial E_c}{\partial T_c} \cdot \Delta T_c - \lambda \frac{\partial E_c}{\partial T_{gs}} \cdot \Delta T_{gs}$$

where $\Delta T_c$ and $\Delta T_{gs}$ denote the change of $T_c$ and $T_{gs}$ during a time step.

For groundcover and soil, Eq. (2) can be written as:

$$C_s \frac{\Delta T_{gs}}{\Delta t} = (R_{ngs} - H_{gs} - \lambda E_{gs}) + (\frac{\partial R_{ngs}}{\partial T_c} \cdot \Delta T_c + \frac{\partial R_{ngs}}{\partial T_{gs}} \cdot \Delta T_{gs})$$

$$- (\frac{\partial H_{gs}}{\partial T_c} \cdot \Delta T_c + \frac{\partial H_{gs}}{\partial T_{gs}} \cdot \Delta T_{gs}) - (\lambda \frac{\partial E_{gs}}{\partial T_c} \cdot \Delta T_c - \lambda \frac{\partial E_{gs}}{\partial T_{gs}} \cdot \Delta T_{gs}) \quad \text{(A2)}$$

$$+ L_{il} \rho_i \frac{\Delta \theta_i}{\Delta t} + 2\lambda_{eff} \frac{T_2 - T_{gs}}{\Delta z_1 + \Delta z_2}$$

For the inner layer, Eq. (3) can be written as:

$$C_s \cdot \Delta z_j \cdot \frac{\Delta T_{s,j}}{\Delta t} = L_{il} \rho_i \frac{\Delta \theta_{i,j}}{\Delta t} \cdot \Delta z_j + 2\lambda_{eff,j} \frac{T_{s,j+1} - T_{s,j}}{\Delta z_j + \Delta z_{j+1}} - 2\lambda_{eff,j-1} \frac{T_{s,j} - T_{s,j-1}}{\Delta z_{j-1} + \Delta z_j} \qquad \text{(A3)}$$

Assuming the total water mass is conserved during one time step, the change of soil temperature ($\Delta T_s$) and the change of soil ice content ($\Delta \theta_i$) can be derived based on Eq. (5):

$$\Delta \theta_i = \frac{(\theta_s)^{-b}(\frac{L_{il}}{g\psi_0 T_f})}{(1+c_k\theta_i)(\theta_T - \theta_i)^{-b}[b(\theta_T - \theta_i)^{-1}(1+c_k\theta_i) + 2c_k]} \Delta T_{gs} \qquad \text{(A4)}$$

Combing Eq. (5) and Eq. (A1)–(A3), the energy balance equation system can be reorganized as a tridiagonal linear equation system with soil temperature of all the soil layers:

$$A_j \cdot \Delta T_{s,j-1} + B_j \cdot \Delta T_{s,j} + C_j \cdot \Delta T_{s,j+1} = D_j \qquad \text{(A5)}$$

where $A_j$, $B_j$ and $C_j$ are the known coefficients and functions of $T_{s,j-1}$, $T_{s,j}$ and $T_{s,j+1}$ at the previous time step. $D_j$ also represents the known values at the previous time.

After solving the tridiagonal matrix for the soil temperature change at different soil layers, we can obtain the soil temperature at the current time step ($T_{s,j}$). In addition, the phase change between liquid water and ice in soil can be decided using the change of soil temperature during one time step. Because

the phase change has been included while solving the temperature tridiagonal matrix, here we can obtain $\Delta \theta_i$ using Eq. (A4). Then we can solve the water fluxes at the interface of the soil layers:

$$Q_{l,j} = K_{l,j}\left[2\frac{\psi_j - \psi_{j+1}}{z_j + z_{j+1}} + 1\right] \qquad \text{(A6)}$$

Combining Eq. (A6) with Eq. (4), we can obtain water balance equations for the surface layer:

$$\psi_1^{k+1} = \psi_1^k - \frac{\Delta t}{z_1}\left(\frac{\partial \psi_1}{\partial \theta_{l,1}} Q_{l,1}\right) - \frac{\Delta t}{z_1}E_{gs} \qquad \text{(A7)}$$

for the root zone layer:

$$\psi_j^{k+1} = \psi_j^k + \frac{\Delta t}{z_j}\frac{\partial \psi_j}{\partial \theta_{l,j}}\left(Q_{l,j-1} - Q_{l,j} - E_{ct}\right) \qquad \text{(A8)}$$





and for the bottom soil layer:

$$\psi_N^{k+1} = \psi_N^k + \frac{\Delta t}{Z_N} \frac{\partial \psi_N}{\partial \theta_{l,N}} (Q_{l,N-1} - q_G) \qquad \text{(A9)}$$

Inserting Eqs. (A7)–(A9) into Eq. (A6) and then regrouping Eq. (A6), then we obtain a tridiagonal linear system for liquid water flow as follows:

$$a_j Q_{j-1} + b_j Q_j + c_j Q_{j+1} = d_j \qquad \text{(A10)}$$

where $a_j$, $b_j$ and $d_j$ are the known coefficients and functions of $\psi_j$, $\psi_{j+1}$ and $\theta_{l,j}$, $\theta_{l,j+1}$ at the previous time step. Therefore, the water fluxes at the current time step can be solved using the above tridiagonal matrix. Then the liquid water content at the current time step can be easily obtained from the following equation:

$$\theta_{l,j}^{k+1} = \theta_{l,j}^k + \frac{\Delta t}{Z_j} (Q_j - Q_{j+1}) \qquad \text{(A11)}$$

**Code and data availability**

The SSiB3-FSM code is available upon request from the first author. The analyses are developed within the Grads and Python software environment. The scripts are also available upon request from the first author. The CAMS gridded 2-m temperature is available at https://www.esrl.noaa.gov/psd/data/gridded/data.ghcncams.html. The Princeton global meteorological forcing data set is available at http://hydrology.princeton.edu/data.php. The China Ground 2-m Temperature Grid Dataset is available at http://cdc.nmic.cn.

**Author contributions**

QL conceived the study and developed the model. QL and YKX developed the framework of the study. QL, YKX analyzed all of the simulation results and wrote the paper. QL and YL drafted the figures. All authors discussed the results and contributed to finalizing the paper.

**Competing interests.**

The authors declare that they have no conflict of interest.

**Acknowledgements**

This research is financially supported by the National Key Research and Development Program of China (2018YFC1505701) and U.S. National Science Foundation AGS-1849654.

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
