# Peer review of "Impact of frozen soil processes on soil thermal characteristics at seasonal to decadal scales over the Tibetan Plateau and North China"

_Hydrology and Earth System Sciences, 2020_

## Referee Comment (RC1) · Anonymous Referee #1 · 28 Nov 2020

GeneralïijŽ Review on manuscript "Impact of frozen soil processes on soil thermal characteristics at seasonal to decadal scales over the Tibetan Plateau and North China" The manuscript addresses frozen soil degradation and surface soil warming issues by introducing a realistic and computationally model which is more stable physically and efficient frozen soil module (FSM) into a land surface model—the third-generation Simplified Simple Biosphere model (SSiB3-FSM) in Tibetan Plateau and North China region. To this end, the performance of the used model, as well as the effects of frozen soil process on the soil temperature profile and soil thermal characteristics, were investigated over the using observation and models simulations. It an intriguing research topic whose rationale has been well established by the authors. The methods seem

more likely to acceptable/reliable, the originality of the research is undoubted. The results interpretation and validation is appropriate and the manuscript is written well. In my opinion, the content of the manuscript fits well to getting published with HESS in its current form due to data availability and above-mentioned qualities. However, I would like to mention some minor concerns which need to be addressed before acceptance. The abstract could have been much improved by mentioning obtained observed and simulated results. There must be a take-home message at the end of the abstract, how the changing climate affected TP and NC concerning frozen soil properties and permafrost? The authors emphasized more on the model used rather than results. They even didn't mention the study period (1981-2005). I would like to know why the decreasing trend of MFD stabilized (line 450-455) after 2000 while glacier mass balance results are in phase with global warming in TP. Most of the glaciers are losing mass and collapsing such as ARU glacier. Does it make sense? Please address this issue.

DetailedïijŽ In the caption of Figure 1, in order to distinguish "The heat and water flux between soil layers are represented by H and Q." with surface sensible heat flux "H" in the Figure 1, " heat flux H" could be changed to "Hk" and "water flux Q" could be changed to "Qk". T in equation (5) should be Ts? In Figure 2, at the last step, the soil temperature, soil ice content and soil liquid water content at k+1 time step should be calculated. So the soil liquid water content $\theta\_(l,j)^k$ should be $\theta\_(l,j)^{(k+1)}$. At line 207, why "nine soil layers over the TP region" were selected? Please clarify it. At line 271, line 306, line 318, please change "(2) Soil temperature profile in the TP", "(3) Soil temperature profile in NC" and "(4) Comparison with the force-restore method" to the corresponding subsection heading. For example, "(2) Soil temperature profile in the TP" should be "4.1.2 Soil temperature profile in the TP". At line 412, it should be 15cm, not "1.5cm".

---

## Referee Comment (RC2) · Anonymous Referee #2 · 13 Dec 2020

The soil freezing and thawing processes are vital for land surface models because they influence both the soil thermal and hydrological variables and modified the land surface memory which play important roles in determining land-atmosphere interactions. However, current land surface models solve the freezing-thawing cycle by using a separated method, where soil temperature is firstly calculated and then soil freezing and thawing processes are adjusted. Li et al., coupled a physically more realistic and computationally more stable and efficient frozen soil module (FSM) into a land surface model the third-generation Simplified Simple Biosphere model (SSiB3-FSM), and evaluated the new model over the Tibetan Plateau and northern China systematically. They also investigated the influences of soil freezing-thawing cycle on the soil temperature

profile, maximum frozen soil depth and soil memory. Generally, the work is important for the land model community and deepens our understanding of the influences of soil freezing-thawing processes. However, some minor revisions are still needed before its publication. 1.The author used the GHCN-CAMS product to evaluate the T2m simulations of SSiB3-FSM. I am confused that whether the temperature forcing from the Princeton is already the T2m? If so, how do you process the temperature forcings as land surface models usually need the forcings at the lowest level of atmosphere model? 2.The soil layer depth of the SSiB3-FSM is 3m and the soil temperature stations over TP are all seasonally frozen ground whose maximum frozen depth is shallower than 3m. However, for some regions such as the western TP, the maximum frozen depth will be deeper than 3m. Will this influence the results in section 4.2.1 which seems to use the whole TP as study region. 3.In section 4.2.2, the author said that land memory in TP is not given because it has been well studied by previous works. While I still suggest to give the results in the supporting information to improve the good performance of SSiB3-FSM. 4.The author compared the observed and simulated normalized maximum frozen depth in section 4.4, and I wonder why do you directly compare the original values? 5.In section 4.1.1 and Table 1, although the author give the evaluation results of T2m at global scale, the detailed results over TP and NC are still needed. 6.The subsections in 4.1 should be revised. Change the "(2) Soil temperature profile in the TP" to "4.1.2 ...", the "(3) Soil ..." to "4.1.3 ..."

---

## Referee Comment (RC3) · Anonymous Referee #3 · 14 Jan 2021

**Review on manuscript "Impact of frozen soil processes on soil thermal characteristics at seasonal to decadal scales over the Tibetan Plateau and North China"**

This work firstly introduced a multi-layer FSM into the SSiB3 to represent the freezing-thawing process and the heat and water transfer in a multi-layer frozen soil (SSiB3-FSM). To overcome the difficulties in achieving stable numerical solutions for frozen soil, a new semi-implicit scheme and a physics-based freezing-thawing scheme were applied to solve the governing equations. With that, the performance of the SSiB3-FSM model, as well as the effects of frozen soil process on the soil temperature profile and soil memory and maximum frozen soil depth, were investigated by using observation and models simulations over the Tibetan Plateau and North China region. The study allows for a better understanding that frozen soil processes are of great importance in controlling surface water and energy balances during the cold season and in cold regions. Furthermore, it also allows for accurate freeze-thaw cycle simulation and frozen soil predictions.

How to solve the highly nonlinear equations in multi-layer frozen soil are the difficulties in land surface model and the climate models. The study has a great potential to provide guidance for future development of frozen soil model in land surface model and climate models. Few published papers focus on the effects of frozen soil processes on soil temperature, soil memory and maximum frozen depth in the TP and NC. This study provides useful information.

The analyses in the paper are well organized and the results are reasonable. The presentation of this article is generally clear. Based on my evaluation for the merit of this paper, I suggest publication of this paper with some minor revisions.

1. In abstract, line 20. Consider clarifying how they are investigated. Observations are mentioned below, please state which observations were used for the comparison.
2. At the end of the abstract, please consider adding an additional sentence to describe the impact of your findings for future research or practical applications.
3. At line 185, "It can effectively produce stable solutions for long-term integrations

with the heat and mass balances." It's too general. Please provide relevant data or citations to support these statements. Otherwise,

4. At line 272, it is not clear why this is not just Section 4.1.2. Consider revising to use the section headings instead of a numbered list. Otherwise, write a sentence to introduce the list first. The same problems are at line 306 and 308.

5. At line 347-349, "To more clearly display these relationships, the soil temperature phase lag time, defined as the point at which the cross-correlation with the first soil layer equals 1, with depth is shown in Fig. 7b and Fig. 8b." Please check that this is what was meant here. Please revise to clarify.

6. Why do you plot Figure 11 with the simulated soil temperature instead of observed data?

7. There might be errors in Figure 2. Please check at the last step of Fig.2 $\theta_{l,j}^{k}$ should be $\theta_{l,j}^{k+1}$ or not.

8. Please check it is "1.5cm" or "15cm" at line 412.

---

## Author Comment (AC1) · 10 Feb 2021

We would like to thank Anonymous Referee #1 (AR1) for their constructive and positive comments. Below, we will respond to the comments made by AR1: the comments from AR1 in black, our response in blue.

The manuscript addresses frozen soil degradation and surface soil warming issues by introducing a realistic and computationally model which is more stable physically and efficient frozen soil module (FSM) into a land surface model—the third-generation Simplified Simple Biosphere model (SSiB3-FSM) in Tibetan Plateau and North China region. To this end, the performance of the used model, as well as the effects of frozen soil process on the soil temperature profile and soil thermal characteristics, were investigated over the using observation and models simulations. It an intriguing research topic whose rationale has been well established by the authors. The methods seem more likely to acceptable/reliable, the originality of the research is undoubted. The results interpretation and validation is appropriate and the manuscript is written well. In my opinion, the content of the manuscript fits well to getting published with HESS in its current form due to data availability and above-mentioned qualities.
However, I would like to mention some minor concerns which need to be addressed before acceptance.
R: Thank you very much for your helpful comments and suggestions for the improvements of the manuscript. We have responded to the following comments or questions and modified the manuscript accordingly.

The abstract could have been much improved by mentioning obtained observed and simulated results. There must be a take-home message at the end of the abstract, how the changing climate affected TP and NC concerning frozen soil properties and permafrost? The authors emphasized more on the model used rather than results. They even didn't mention the study period (1981-2005).
R: To better clarify the impact of frozen soil process on TP and NC and provide a take-home message for the readers, we modified the abstract by adding description of the study results at the end of the abstract. See line 30-31. And we also emphasized the study period is from 1981-2005. See line 20-21.

I would like to know why the decreasing trend of MFD stabilized (line 450-455) after 2000 while glacier mass balance results are in phase with global warming in TP. Most of the glaciers are losing mass and collapsing such as ARU glacier. Does it make sense? Please address this issue.
R: Thanks for your suggestions. Large number of studies have shown the annual soil temperature over TP has been increasing since 1980s and the rate is even more pronounced than the global warming (Liu and Chen, 2000). Therefore, the TP glaciers experience abrupt retreat under climate warming with westerly monsoon interaction (Yao et al. 2012). However, MFD was mainly controlled by the winter surface temperature. Spatio-temporal analysis of surface temperature over the TP during 1981-2015 shows that the winter surface temperature over the TP increased significantly in the 1980s, and the temperature changes were relatively stable in the 1990s and early 21st century (Bai et al., 2018). That is why the decreasing trend of MFD over TP after 2000 is stabilized. As for the variation of MFD after 2000, we add above discussion in the manuscript. See line 487-491.

References:

Liu X.D., Chen B. D., 2000, Climatic warming in the Tibetan Plateau during recent decades. International Journal of Climatology, 20 (14): 1729-1742.

Bai Lu, Yao Y. B., Lei X.X, Zhang L., 2018, Annual and seasonal variation characteristics of Surface temperature in the Tibetan Plateau in recent 40 years. Journal of Geomatics, 43(2): 15-18.

Yao T.D., and Coauthors, 2012: Different glacier status with atmospheric circulations in Tibetan Plateau and surroundings. Nat. Climate Change, 2, 663–667, https://doi.org/10.1038/nclimate1580.

In the caption of Figure 1, in order to distinguish "The heat and water flux between soil layers are represented by $H$ and $Q$." with surface sensible heat flux "$H$" in the Figure 1, " heat flux $H$" could be changed to "$H_k$" and "water flux $Q$" could be changed to "$Q_k^{''}$".

R: Agreed. We have corrected this in Figure 1.

$T$ in equation (5) should be $T_s$?

R: Yes. It should be $T_s$ in equation (5) and we have corrected it.

In Figure 2, at the last step, the soil temperature, soil ice content and soil liquid water content at $k+1$ time step should be calculated. So the soil liquid water content $\theta_{l,j}^{k}$ should be $\theta_{l,j}^{k+1}$.

R: Agreed. We have corrected it in Figure 2.

At line 207, why "nine soil layers over the TP region" were selected? Please clarify it.

R: Because the soil temperature are observed at nine soil layers. They are at 0, 5, 10, 15, 20, 40, 80, 160 and 320 cm. The information about observed nine soil layers can be found at line 213-214.

At line 271, line 306, line 318, please change "(2) Soil temperature profile in the TP", "(3) Soil temperature profile in NC" and "(4) Comparison with the force-restore method" to the corresponding subsection heading. For example, "**(2) Soil temperature profile in the TP**" should be "**4.1.2 Soil temperature profile in the TP**".

R: Agreed. We have revised them.

At line 412, it should be 15cm, not "1.5cm".

R: Yes. It should be 1.5cm. We have corrected it. See line 428.

---

## Author Comment (AC2) · 10 Feb 2021

We would like to thank Anonymous Referee #2 (AR2) for their constructive and positive comments. Below, we will respond to the comments made by AR2: the comments from AR2 in black, our response in blue.

The soil freezing and thawing processes are vital for land surface models because they influence both the soil thermal and hydrological variables and modified the land surface memory which play important roles in determining land-atmosphere interactions. However, current land surface models solve the freezing-thawing cycle by using a separated method, where soil temperature is firstly calculated and then soil freezing and thawing processes are adjusted. Li et al., coupled a physically more realistic and computationally more stable and efficient frozen soil module (FSM) into a land surface model the third-generation Simplified Simple Biosphere model (SSiB3-FSM), and evaluated the new model over the Tibetan Plateau and northern China systematically. They also investigated the influences of soil freezing-thawing cycle on the soil temperature profile, maximum frozen soil depth and soil memory. Generally, the work is important for the land model community and deepens our understanding of the influences of soil freezing-thawing processes. However, some minor revisions are still needed before its publication.

We thank the reviewer for a thorough and knowledgeable reading of the paper and your helpful comments and suggestions for the improvements of the manuscript. We have responded to the following comments or questions and modified the manuscript accordingly.

1. The author used the GHCN-CAMS product to evaluate the T2m simulations of SSiB3-FSM. I am confused that whether the temperature forcing from the Princeton is already the T2m? If so, how do you process the temperature forcings as land surface models usually need the forcings at the lowest level of atmosphere model?

R: It is true that surface air temperature is included in Princeton forcing data and is used to drive the SSiB3-FSM and SSiB3 in this study with other forcing variables (e.g. radiation, precipitation, wind and humidity). Princeton forcing data set provides a long-term, globally consistent data set of near-surface meteorological variables that can be used to drive models of the terrestrial hydrologic and ecological processes for the study of seasonal and interannual variability. This data set is constructed by combining a suite of global observation-based datasets with the National Centers for Environmental Prediction–National Center for Atmospheric Research (NCEP–NCAR) reanalysis. It has been widely used by many model studies. In this study, we aimed to validate the land surface skin temperature (not T2m) simulated by SSiB3-FSM and SSiB3 globally. So GHCN-CAMS surface air temperature data, which is a station observation based global monthly land surface air temperature data and was developed at the Climate Prediction Center, National Centers for Environmental Prediction, was used to validate the models in view of its superiority over other data sets over global domains. Although GHCN-CAMS data is air temperature data, in fact the changes in surface air temperature are highly consistent with those in skin temperature. Therefore, in this study Princeton forcing data was used to drive the models and

GHCN-CAMS air temperature data was used to validate the simulated land surface skin temperature globally. We added related information at line 207-210.

2. The soil layer depth of the SSiB3-FSM is 3m and the soil temperature stations over TP are all seasonally frozen ground whose maximum frozen depth is shallower than 3m. However, for some regions such as the western TP, the maximum frozen depth will be deeper than 3m. Will this influence the results in section 4.2.1 which seems to use the whole TP as study region.

R: The deepest soil column depth in SSiB3-FSM is not only 3m. The soil column in model is discretized into eight, eleven and twelve layers for desert, grassland and trees, respectively. And the deepest soil depth is also depends on the upper vegetation type. For example, the deepest soil depth over bare soil and grassland is 7.77 m and over forest it is about 12m. They are much larger than 3m. In this study, over TP we just focus on the 14 sites (Figure 1). They are almost located in the eastern part of TP, so the MFD is also shallower than western TP. We added the information about soil depth at line 128-130.

3. In section 4.2.2, the author said that land memory in TP is not given because it has been well studied by previous works. While I still suggest to give the results in the supporting information to improve the good performance of SSiB3-FSM.

R: Thanks for the suggestion. We added the land memory in TP in Table 3 in section 4.2.2.

4. The author compared the observed and simulated normalized maximum frozen depth in section 4.4, and I wonder why do you directly compare the original values?

R: We added the reason why we compared the normalized MFD in section 4.4. See line 458-460. Because there is a large difference between the original values for MFD in simulated results and the observation data, the zero-score normalization method was used to process the observed and simulated MFD. The difference of MFD between simulation and observation may come from the systematic bias between forcing data and observation. This bias is unavoidable and it does not affect the variation or trend of soil temperature, just shown in Figure 4 and Figure 5. Therefore, the zero-score normalization method is used in order to analysis and compare the variation and the trend of MFD both by simulation and observation.

5. In section 4.1.1 and Table 1, although the author give the evaluation results of T2m at global scale, the detailed results over TP and NC are still needed.

R: We gave the detailed results over TP and NC in section 4.1.1 and added them in Table 1.

6. The subsections in 4.1 should be revised. Change the "(2) Soil temperature profile in the TP" to "4.1.2 ...", the "(3) Soil ..." to "4.1.3 ...".

R: Yes. We have revised them.

---

## Author Comment (AC3) · 10 Feb 2021

We would like to thank Anonymous Referee #3 (AR3) for their constructive and positive comments. Below, we will respond to the comments made by AR3: the comments from AR3 in black, our response in blue.

This work firstly introduced a multi-layer FSM into the SSiB3 to represent the freezing-thawing process and the heat and water transfer in a multi-layer frozen soil (SSiB3-FSM). To overcome the difficulties in achieving stable numerical solutions for frozen soil, a new semi-implicit scheme and a physics-based freezing-thawing scheme were applied to solve the governing equations. With that, the performance of the SSiB3-FSM model, as well as the effects of frozen soil process on the soil temperature profile and soil memory and maximum frozen soil depth, were investigated over the using observation and models simulations over the Tibetan Plateau and North China region. The study allows for a better understanding that frozen soil processes are of great importance in controlling surface water and energy balances during the cold season and in cold regions. Furthermore, it also allows for accurate freeze-thaw cycle simulation and frozen soil predictions.

How to solve the highly nonlinear equations in multi-layer frozen soil are the difficulties in land surface model and the climate models. The study has a great potential to provide guidance for future development of frozen soil model in land surface model and climate models. Few published papers focus on the effects of frozen soil processes on soil temperature, soil memory and maximum frozen depth in the TP and NC. This study provides useful information.

The analyses in the paper are well organized and the results are reasonable. The presentation of this article is generally clear. Based on my evaluation for the merit of this paper, I suggest publication of this paper with some revisions.

R: Thank you very much for your helpful comments and suggestions for the improvements of the manuscript. We have responded to the following comments or questions and modified the manuscript accordingly.

1. In abstract, line 20. Consider clarifying how they are investigated. Observations are mentioned below, please state which observations were used for the comparison.

   R: The observation data used to validate the models are mentioned in the abstract. See line

20-21.

2. At the end of the abstract, please consider adding an additional sentence to describe the impact of your findings for future research or practical applications.

R: We modified the abstract by adding description of the study results at the end of the abstract. See line 30-31.

3. At line 185, "It can effectively produce stable solutions for long-term integrations with the heat and mass balances." It's too general. Please provide relevant data or citations to support these statements.

R: We rewrote this sentence. See line 188-191.

4. At line 272, it is not clear why this is not just Section 4.1.2. Consider revising to use the section headings instead of a numbered list. Otherwise, write a sentence to introduce the list first. The same problems are at line 306 and 308.

R: Agreed. We have revised them.

5. At line 347-349, "To more clearly display these relationships, the soil temperature phase lag time, defined as the point at which the cross-correlation with the first soil layer equals 1, with depth is shown in Fig. 7b and Fig. 8b." Please check that this is what was meant here. Please revise to clarify.

R: This sentence clarified how we defined the soil temperature phase lag time based on Figure 7a and 8a. We revised this sentence and see line 356-359.

6. Why do you plot Figure 11 with the simulated soil temperature instead of observed data?

R: The observed soil temperature data is monthly data, but the simulated soil temperature data is daily. In order to clearly exhibit the temporal characteristics of soil temperature profile, we used the simulated daily data to plot the climatology of annual soil temperature variation with soil depth over TP and NC. The observed climatology of simulated daily soil temperature (Left) over the TP (14 sites averaged) and (Right) NC (16 sites averaged) for 1981−2005 has also been shown as following. It has shown the similar characteristics with Figure 11.

[Figure]

Figure. The climatology of observed monthly soil temperature (Left) over TP (14 sites averaged) and (Right) North China (16 sites averaged) for the period of 1981-2005. (Unit: ℃)

7. There might be errors in Figure 2. Please check at the last step of Fig.2 $\theta_{l,j}^{k}$ should be $\theta_{l,j}^{k+1}$ or not.

    R: Agreed. We have corrected it.

8. Please check it is "1.5cm" or "15cm" at line 412.

    R: Yes. It should be 1.5 cm. We have revised it.